# Movement Pruning:
# Adaptive Sparsity by Fine-Tuning

**Victor Sanh[1], Thomas Wolf[1], Alexander M. Rush[1,2]**
[1]Hugging Face, [2]Cornell University
{victor,thomas}@huggingface.co ; arush@cornell.edu

## Abstract

Magnitude pruning is a widely used strategy for reducing model size in pure supervised learning; however, it is less effective in the transfer learning regime that has become standard for state-of-the-art natural language processing applications. We propose the use of *movement pruning*, a simple, deterministic first-order weight pruning method that is more adaptive to pretrained model fine-tuning. We give mathematical foundations to the method and compare it to existing zeroth- and first-order pruning methods. Experiments show that when pruning large pretrained language models, movement pruning shows significant improvements in high-sparsity regimes. When combined with distillation, the approach achieves minimal accuracy loss with down to only 3% of the model parameters.

## 1 Introduction

Large-scale transfer learning has become ubiquitous in deep learning and achieves state-of-the-art performance in applications in natural language processing and related fields. In this setup, a large model pretrained on a massive generic dataset is then fine-tuned on a smaller annotated dataset to perform a specific end-task. Model accuracy has been shown to scale with the pretrained model and dataset size [Raffel et al., 2019]. However, significant resources are required to ship and deploy these large models, and training the models have high environmental costs [Strubell et al., 2019].

Sparsity induction is a widely used approach to reduce the memory footprint of neural networks at only a small cost of accuracy. Pruning methods, which remove weights based on their importance, are a particularly simple and effective method for compressing models. Smaller models are easier to sent on edge devices such as mobile phones but are also significantly less energy greedy: the majority of the energy consumption comes from fetching the model parameters from the long term storage of the mobile device to its volatile memory [Han et al., 2016a, Horowitz, 2014].

Magnitude pruning [Han et al., 2015, 2016b], which preserves weights with high absolute values, is the most widely used method for weight pruning. It has been applied to a large variety of architectures in computer vision [Guo et al., 2016], in language processing [Gale et al., 2019], and more recently has been leveraged as a core component in the *lottery ticket hypothesis* [Frankle et al., 2020].

While magnitude pruning is highly effective for standard supervised learning, it is inherently less useful in the transfer learning regime. In supervised learning, weight values are primarily determined by the end-task training data. In transfer learning, weight values are mostly predetermined by the original model and are only fine-tuned on the end task. This prevents these methods from learning to prune based on the fine-tuning step, or "fine-pruning."

In this work, we argue that to effectively reduce the size of models for transfer learning, one should instead use *movement pruning*, i.e., pruning approaches that consider the changes in weights during fine-tuning. Movement pruning differs from magnitude pruning in that both weights with low and high values can be pruned if they shrink during training. This strategy moves the selection criteria

from the 0th to the 1st-order and facilitates greater pruning based on the fine-tuning objective. To test this approach, we introduce a particularly simple, deterministic version of movement pruning utilizing the straight-through estimator [Bengio et al., 2013].

We apply movement pruning to pretrained language representations (BERT) [Devlin et al., 2019, Vaswani et al., 2017] on a diverse set of fine-tuning tasks. In highly sparse regimes (less than 15% of remaining weights), we observe significant improvements over magnitude pruning and other 1st-order methods such as $L_0$ regularization [Louizos et al., 2017]. Our models reach 95% of the original BERT performance with only 5% of the encoder's weight on natural language inference (MNLI) [Williams et al., 2018] and question answering (SQuAD v1.1) [Rajpurkar et al., 2016]. Analysis of the differences between magnitude pruning and movement pruning shows that the two methods lead to radically different pruned models with movement pruning showing greater ability to adapt to the end-task.

## 2 Related Work

In addition to magnitude pruning, there are many other approaches for generic model weight pruning. Most similar to our approach are methods for using parallel score matrices to augment the weight matrices [Mallya and Lazebnik, 2018, Ramanujan et al., 2020], which have been applied for convolutional networks. Differing from our methods, these methods keep the weights of the model fixed (either from a randomly initialized network or a pre-trained network) and the scores are updated to find a good sparse subnetwork.

Many previous works have also explored using higher-order information to select prunable weights. LeCun et al. [1989] and Hassibi et al. [1993] leverage the Hessian of the loss to select weights for deletion. Our method does not require the (possibly costly) computation of second-order derivatives since the importance scores are obtained simply as the by-product of the standard fine-tuning. [Theis et al., 2018, Ding et al., 2019, Lee et al., 2019] use the absolute value or the square value of the gradient. In contrast, we found it useful to preserve the direction of movement in our algorithm.

Compressing pretrained language models for transfer learning is also a popular area of study. Other approaches include knowledge distillation [Sanh et al., 2019, Tang et al., 2019] and structured pruning [Fan et al., 2020a, Sajjad et al., 2020, Michel et al., 2019, Wang et al., 2019]. Our core method does not require an external teacher model and targets individual weight. We also show that having a teacher can further improve our approach. Recent work also builds upon iterative magnitude pruning with rewinding [Yu et al., 2020], weight redistribution [Dettmers and Zettlemoyer, 2019] models from scratch. This differs from our approach which we frame in the context of transfer learning (focusing on the fine-tuning stage). Finally, another popular compression approach is quantization. Quantization has been applied to a variety of modern large architectures [Fan et al., 2020b, Zafrir et al., 2019, Gong et al., 2014] providing high memory compression rates at the cost of no or little performance. As shown in previous works [Li et al., 2020, Han et al., 2016b] quantization and pruning are complimentary and can be combined to further improve the performance/size ratio.

## 3 Background: Score-Based Pruning

We first establish shared notation for discussing different neural network pruning strategies. Let $\mathbf{W} \in \mathbb{R}^{n \times n}$ refer to a generic weight matrix in the model (we consider square matrices, but they could be of any shape). To determine which weights are pruned, we introduce a parallel matrix of associated importance scores $\mathbf{S} \in \mathbb{R}^{n \times n}$. Given importance scores, each pruning strategy computes a mask $\mathbf{M} \in \{0, 1\}^{n \times n}$. Inference for an input $\mathbf{x}$ becomes $\mathbf{a} = (\mathbf{W} \odot \mathbf{M})\mathbf{x}$, where $\odot$ is the Hadamard product. A common strategy is to keep the top-$v$ percent of weights by importance. We define $\text{Top}_v$ as a function which selects the $v\%$ highest values in $\mathbf{S}$:

$$\text{Top}_v(\mathbf{S})_{i,j} = \begin{cases} 1, & S_{i,j} \text{ in top } v\% \\ 0, & \text{o.w.} \end{cases} \tag{1}$$

Magnitude-based weight pruning determines the mask based on the absolute value of each weight as a measure of importance. Formally, we have importance scores $\mathbf{S} = \left(|W_{i,j}|\right)_{1 \leq i,j \leq n}$, and masks $\mathbf{M} = \text{Top}_v(\mathbf{S})$ (Eq (1)). There are several extensions to this base setup. Han et al. [2015] use

| | Magnitude pruning | $L_0$ regularization | Movement pruning | Soft movement pruning |
|---|---|---|---|---|
| Pruning Decision | 0th order | 1st order | 1st order | 1st order |
| Masking Function | $\text{Top}_v$ | Continuous Hard-Concrete | $\text{Top}_v$ | Thresholding |
| Pruning Structure | Local or Global | Global | Local or Global | Global |
| Learning Objective | $\mathcal{L}$ | $\mathcal{L} + \lambda_{l0}\mathbb{E}(L_0)$ | $\mathcal{L}$ | $\mathcal{L} + \lambda_{\text{mvp}}R(\mathbf{S})$ |
| Gradient Form | | Gumbel-Softmax | Straight-Through | Straight-Through |
| Scores $\mathbf{S}$ | $|W_{i,j}|$ | $-\sum_t(\frac{\partial\mathcal{L}}{\partial W_{i,j}})^{(t)}W_{i,j}^{(t)}f(\overline{S}_{i,j}^{(t)})$ | $-\sum_t(\frac{\partial\mathcal{L}}{\partial W_{i,j}})^{(t)}W_{i,j}^{(t)}$ | $-\sum_t(\frac{\partial\mathcal{L}}{\partial W_{i,j}})^{(t)}W_{i,j}^{(t)}$ |

Table 1: Summary of the pruning methods considered in this work and their specificities. The expression of $f$ of $L_0$ regularization is detailed in Eq (4).

iterative magnitude pruning: the model is first trained until convergence and weights with the lowest magnitudes are removed afterward. The sparsified model is then re-trained with the removed weights fixed to 0. This loop is repeated until the desired sparsity level is reached.

In this study, we focus on *automated gradual pruning* [Zhu and Gupta, 2018]. It supplements magnitude pruning by allowing masked weights to be updated such that they are not fixed for the entire duration of the training. Automated gradual pruning enables the model to recover from previous masking choices [Guo et al., 2016]. In addition, one can gradually increases the sparsity level $v$ during training using a cubic sparsity scheduler: $v^{(t)} = v_f + (v_i - v_f)\left(1 - \frac{t-t_i}{N\Delta t}\right)^3$. The sparsity level at time step $t$, $v^{(t)}$ is increased from an initial value $v_i$ (usually 0) to a final value $v_f$ in $n$ pruning steps after $t_i$ steps of warm-up. The model is thus pruned and trained jointly.

## 4   Movement Pruning

Magnitude pruning can be seen as utilizing zeroth-order information (absolute value) of the running model. In this work, we focus on movement pruning methods where importance is derived from first-order information. Intuitively, instead of selecting weights that are far from zero, we retain connections that are moving away from zero during the training process. We consider two versions of movement pruning: hard and soft.

For (hard) movement pruning, masks are computed using the $\text{Top}_v$ function: $\mathbf{M} = \text{Top}_v(\mathbf{S})$. Unlike magnitude pruning, during training, we learn both the weights $\mathbf{W}$ and their importance scores $\mathbf{S}$. During the forward pass, we compute for all $i$, $a_i = \sum_{k=1}^n W_{i,k}M_{i,k}x_k$.

Since the gradient of $\text{Top}_v$ is 0 everywhere it is defined, we follow Ramanujan et al. [2020], Mallya and Lazebnik [2018] and approximate its value with the *straight-through estimator* [Bengio et al., 2013]. In the backward pass, $\text{Top}_v$ is ignored and the gradient goes "straight-through" to $\mathbf{S}$. The approximation of gradient of the loss $\mathcal{L}$ with respect to $S_{i,j}$ is given by

$$\frac{\partial\mathcal{L}}{\partial S_{i,j}} = \frac{\partial\mathcal{L}}{\partial a_i}\frac{\partial a_i}{\partial S_{i,j}} = \frac{\partial\mathcal{L}}{\partial a_i}W_{i,j}x_j \tag{2}$$

This implies that the scores of weights are updated, even if these weights are masked in the forward pass. We prove in Appendix A.1 that movement pruning as an optimization problem will converge.

We also consider a relaxed (soft) version of movement pruning based on the binary mask function described by Mallya and Lazebnik [2018]. Here we replace hyperparameter $v$ with a fixed global threshold value $\tau$ that controls the binary mask. The mask is calculated as $\mathbf{M} = (\mathbf{S} > \tau)$. In order to control the sparsity level, we add a regularization term $R(\mathbf{S}) = \lambda_{\text{mvp}}\sum_{i,j}\sigma(S_{i,j})$ which encourages the importance scores to decrease over time[1]. The coefficient $\lambda_{\text{mvp}}$ controls the penalty intensity and thus the sparsity level.

**Method Interpretation**   In movement pruning, the gradient of $\mathcal{L}$ with respect to $W_{i,j}$ is given by the standard gradient derivation: $\frac{\partial\mathcal{L}}{\partial W_{i,j}} = \frac{\partial\mathcal{L}}{\partial a_i}M_{i,j}x_j$. By combining it to Eq (2), we have $\frac{\partial\mathcal{L}}{\partial S_{i,j}} = \frac{\partial\mathcal{L}}{\partial W_{i,j}}W_{i,j}$ (we omit the binary mask term $M_{i,j}$ for simplicity). From the gradient update in Eq (2), $S_{i,j}$ is increasing when $\frac{\partial\mathcal{L}}{\partial S_{i,j}} < 0$, which happens in two cases:

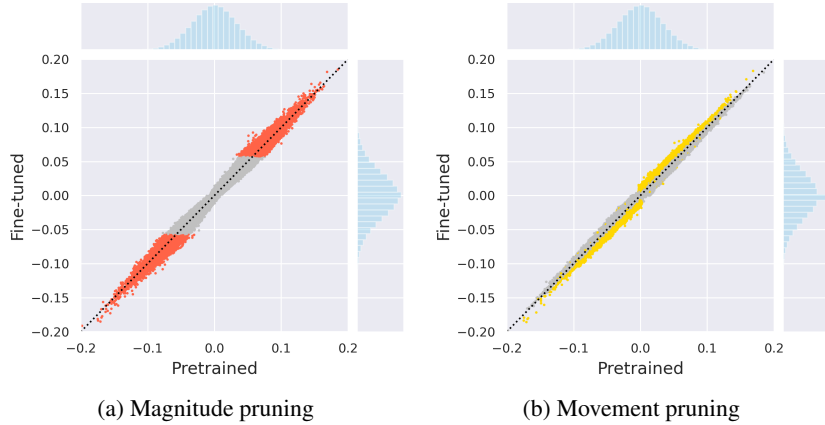

(a) Magnitude pruning        (b) Movement pruning

Figure 1: During fine-tuning (on MNLI), the weights stay close to their pre-trained values which limits the adaptivity of magnitude pruning. We plot the identity line in black. Pruned weights are plotted in grey. Magnitude pruning selects weights that are far from 0 while movement pruning selects weights that are moving away from 0.

- (a) $\frac{\partial \mathcal{L}}{\partial W_{i,j}} < 0$ and $W_{i,j} > 0$
- (b) $\frac{\partial \mathcal{L}}{\partial W_{i,j}} > 0$ and $W_{i,j} < 0$

It means that during training $W_{i,j}$ is increasing while being positive or is decreasing while being negative. It is equivalent to saying that $S_{i,j}$ is increasing when $W_{i,j}$ is moving away from 0. Inversely, $S_{i,j}$ is decreasing when $\frac{\partial \mathcal{L}}{\partial S_{i,j}} > 0$ which means that $W_{i,j}$ is shrinking towards 0.

While magnitude pruning selects the most important weights as the ones which maximize their distance to 0 ($|W_{i,j}|$), movement pruning selects the weights which are moving the most away from 0 ($S_{i,j}$). For this reason, magnitude pruning can be seen as a 0th order method, whereas movement pruning is based on a 1st order signal. In fact, $\mathbf{S}$ can be seen as an accumulator of movement: from equation (2), after $T$ gradient updates, we have

$$S_{i,j}^{(T)} = -\alpha_S \sum_{t<T} (\frac{\partial \mathcal{L}}{\partial W_{i,j}})^{(t)} W_{i,j}^{(t)} \qquad (3)$$

Figure 1 shows this difference empirically by comparing weight values during fine-tuning against their pre-trained value. As observed by Gordon et al. [2020], fine-tuned weights stay close in absolute value to their initial pre-trained values. For magnitude pruning, this stability around the pre-trained values implies that we know with high confidence before even fine-tuning which weights will be pruned as the weights with the smallest absolute value at pre-training will likely stay small and be pruned. In contrast, in movement pruning, the pre-trained weights do not have such an awareness of the pruning decision since the selection is made during fine-tuning (moving away from 0), and both low and high values can be pruned. We posit that this is critical for the success of the approach as it is able to prune based on the task-specific data, not only the pre-trained value.

**$L_0$ Regularization**    Finally we note that movement pruning (and its soft variant) yield a similar update as $L_0$ regularization based pruning, another movement based pruning approach [Louizos et al., 2017]. Instead of straight-through, $L_0$ uses the *hard-concrete* distribution, where the mask $\mathbf{M}$ is sampled for all $i, j$ with hyperparameters $b > 0, l < 0,$ and $r > 1$:

$$u \sim \mathcal{U}(0,1) \qquad\qquad \overline{S}_{i,j} = \sigma\big((\log(u) - \log(1-u) + S_{i,j})/b\big)$$
$$Z_{i,j} = (r-l)\overline{S}_{i,j} + l \qquad\qquad M_{i,j} = \min(1, \mathrm{ReLU}(Z_{i,j}))$$

The expected $L_0$ norm has a closed form involving the parameters of the hard-concrete: $\mathbb{E}(L_0) = \sum_{i,j} \sigma\big(\log S_{i,j} - b\log(-l/r)\big)$. Thus, the weights and scores of the model can be optimized in

an end-to-end fashion to minimize the sum of the training loss $\mathcal{L}$ and the expected $L_0$ penalty. A coefficient $\lambda_{l0}$ controls the $L_0$ penalty and indirectly the sparsity level. Gradients take a similar form:

$$\frac{\partial \mathcal{L}}{\partial S_{i,j}} = \frac{\partial \mathcal{L}}{\partial a_i} W_{i,j} x_j f(\overline{S}_{i,j}) \text{ where } f(\overline{S}_{i,j}) = \frac{r-l}{b} \bar{S}_{i,j}(1 - \bar{S}_{i,j}) \mathbf{1}_{\{0 \le Z_{i,j} \le 1\}} \tag{4}$$

At test time, a non-stochastic estimation of the mask is used: $\hat{\mathbf{M}} = \min\left(1, \text{ReLU}\left((r-l)\sigma(\mathbf{S}) + l\right)\right)$ and weights multiplied by 0 can simply be discarded.

Table 1 highlights the characteristics of each pruning method. The main differences are in the masking functions, pruning structure, and the final gradient form.

## 5 Experimental Setup

Transfer learning for NLP uses large pre-trained language models that are fine-tuned on target tasks [Ruder et al., 2019, Devlin et al., 2019, Radford et al., 2019, Liu et al., 2019]. We experiment with task-specific pruning of `BERT-base-uncased`, a pre-trained model that contains roughly 84M parameters. We freeze the embedding modules and fine-tune the transformer layers and the task-specific head. All reported sparsity percentages are relative to `BERT-base` and correspond exactly to model size even comparing to baselines.

We perform experiments on three monolingual (English) tasks, which are common benchmarks for the recent progress in transfer learning for NLP: question answering (SQuAD v1.1) [Rajpurkar et al., 2016], natural language inference (MNLI) [Williams et al., 2018], and sentence similarity (QQP) [Iyer et al., 2017]. The datasets respectively contain 8K, 393K, and 364K training examples. SQuAD is formulated as a span extraction task, MNLI and QQP are paired sentence classification tasks.

For a given task, we fine-tune the pre-trained model for the same number of updates (between 6 and 10 epochs) across pruning methods[2]. We follow Zhu and Gupta [2018] and use a cubic sparsity scheduling for Magnitude Pruning (MaP), Movement Pruning (MvP), and Soft Movement Pruning (SMvP). Adding a few steps of cool-down at the end of pruning empirically improves the performance especially in high sparsity regimes. The schedule for $v$ is:

$$\begin{cases} v_i & 0 \le t < t_i \\ v_f + (v_i - v_f)(1 - \frac{t - t_i - t_f}{N\Delta t})^3 & t_i \le t < T - t_f \\ v_f & \text{o.w.} \end{cases} \tag{5}$$

where $t_f$ is the number of cool-down steps.

We compare our results against several state-of-the-art pruning baselines: Reweighted Proximal Pruning (RPP) [Guo et al., 2019] combines re-weighted $L_1$ minimization and Proximal Projection [Parikh and Boyd, 2014] to perform unstructured pruning. LayerDrop [Fan et al., 2020a] leverages structured dropout to prune models at test time. For RPP and LayerDrop, we report results from authors. We also compare our method against the mini-BERT models, a collection of smaller BERT models with varying hyper-parameters [Turc et al., 2019].

Finally, Gordon et al. [2020], Li et al. [2020] apply unstructured magnitude pruning as a post-hoc operation whereas we use *automated gradual pruning* [Zhu and Gupta, 2018] which improves on these methods by enabling masked weights to be updated. Moreover, McCarley [2019] compares multiple methods to compute structured masking ($L_0$ regularization and head importance as described in [Michel et al., 2019]) and found that structured $L_0$ regularization performs best. We did not find any implementation for this work, so for fair comparison, we presented a strong unstructured $L_0$ regularization baseline.

## 6 Results

Figure 2 displays the results for the main pruning methods at different levels of pruning on each dataset. First, we observe the consistency of the comparison between magnitude and movement

Figure 2: Comparisons between different pruning methods in high sparsity regimes. **Soft movement pruning consistently outperforms other methods in high sparsity regimes.** We plot the performance of the standard fine-tuned BERT along with 95% of its performance. Sparsity percentages are relative to BERT-base and correspond exactly to model size.

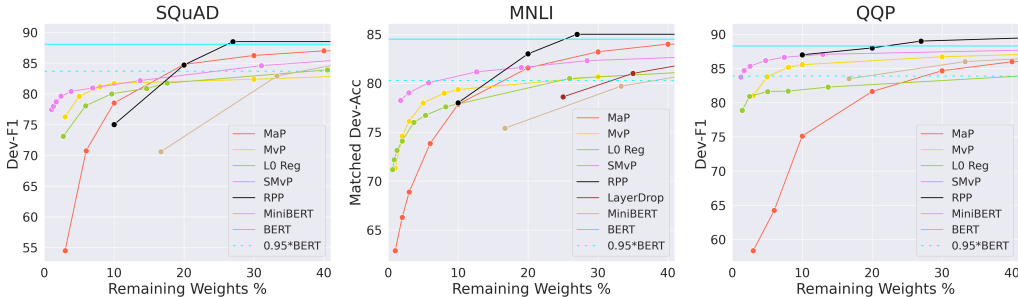

Table 2: Performance at high sparsity levels. **(Soft) movement pruning outperforms current state-of-the art pruning methods at different high sparsity levels.** 3% corresponds to 2.6 millions (M) non-zero parameters in the encoder and 10% to 8.5M.

|  | BERT base fine-tuned | Remaining Weights (%) | MaP | $L_0$ Regu | MvP | soft MvP |
|---|---|---|---|---|---|---|
| SQuAD - Dev EM/F1 | 80.4/88.1 | 10% | 67.7/78.5 | 69.9/80.0 | **71.9/81.7** | 71.3/81.5 |
|  |  | 3% | 40.1/54.5 | 61.2/73.3 | 65.2/76.3 | **69.5/79.9** |
| MNLI - Dev acc/MM acc | 84.5/84.9 | 10% | 77.8/79.0 | 77.9/78.5 | 79.3/79.5 | **80.7/81.1** |
|  |  | 3% | 68.9/69.8 | 75.1/75.4 | 76.1/76.7 | **79.0/79.6** |
| QQP - Dev acc/F1 | 91.4/88.4 | 10% | 78.8/75.1 | 87.5/81.9 | 89.1/85.5 | **90.5/87.1** |
|  |  | 3% | 72.1/58.4 | 86.5/81.0 | 85.6/81.0 | **89.3/85.6** |

pruning: at low sparsity (more than 70% of remaining weights), magnitude pruning outperforms all methods with little or no loss with respect to the dense model whereas the performance of movement pruning methods quickly decreases even for low sparsity levels. However, magnitude pruning performs poorly with high sparsity, and the performance drops extremely quickly. In contrast, first-order methods show strong performances with less than 15% of remaining weights.

Table 2 shows the specific model scores for different methods at high sparsity levels. Magnitude pruning on SQuAD achieves 54.5 F1 with 3% of the weights compared to 73.3 F1 with $L_0$ regularization, 76.3 F1 for movement pruning, and 79.9 F1 with soft movement pruning. These experiments indicate that in high sparsity regimes, importance scores derived from the movement accumulated during fine-tuning induce significantly better pruned models compared to absolute values.

Next, we compare the difference in performance between first-order methods. We see that straight-through based hard movement pruning (MvP) is comparable with $L_0$ regularization (with a significant gap in favor of movement pruning on QQP). Soft movement pruning (SMvP) consistently outperforms hard movement pruning and $L_0$ regularization by a strong margin and yields the strongest performance among all pruning methods in high sparsity regimes. These comparisons support the fact that even if movement pruning (and its relaxed version soft movement pruning) is simpler than $L_0$ regularization, it yet yields stronger performances for the same compute budget.

Finally, movement pruning and soft movement pruning compare favorably to the other baselines, except for QQP where RPP is on par with soft movement pruning. Movement pruning also outperforms the fine-tuned mini-BERT models. This is coherent with [Li et al., 2020]: it is both more efficient and more effective to train a large model and compress it afterward than training a smaller model from scratch. We do note though that current hardware does not support optimized inference for sparse models: from an inference speed perspective, it might often desirable to use a small dense model such as mini-BERT over a sparse alternative of the same size.

**Distillation further boosts performance** Following previous work, we can further leverage knowledge distillation [Bucila et al., 2006, Hinton et al., 2014] to boost performance for free in the pruned domain [Jiao et al., 2019, Sanh et al., 2019] using our baseline fine-tuned BERT-base model as

Figure 3: Comparisons between different pruning methods augmented with distillation. **Distillation improves the performance across all pruning methods and sparsity levels.**

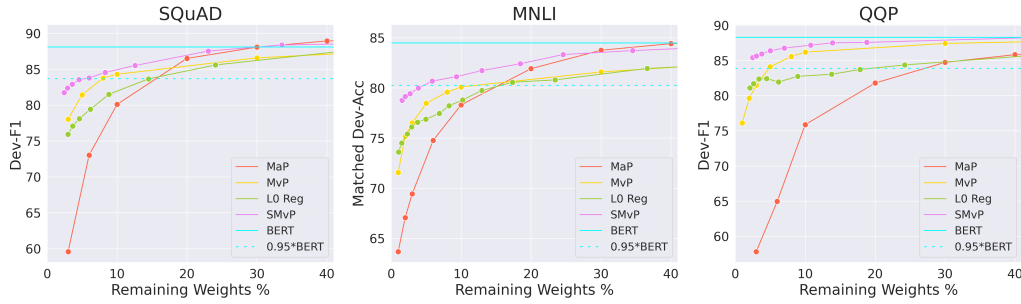

Table 3: Distillation-augmented performances for selected high sparsity levels. **All pruning methods benefit from distillation signal further enhancing the ratio Performance VS Model Size.**

|  | BERT base fine-tuned | Remaining Weights (%) | MaP | $L_0$ Regu | MvP | soft MvP |
|---|---|---|---|---|---|---|
| SQuAD - Dev EM/F1 | 80.4/88.1 | 10% | 70.2/80.1 | 72.4/81.9 | 75.6/84.3 | **76.6/84.9** |
|  |  | 3% | 45.5/59.6 | 64.3/75.8 | 67.5/78.0 | **72.7/82.3** |
| MNLI - Dev acc/MM acc | 84.5/84.9 | 10% | 78.3/79.3 | 78.7/79.7 | 80.1/80.4 | **81.2/81.8** |
|  |  | 3% | 69.4/70.6 | 76.0/76.2 | 76.5/77.4 | **79.5/80.1** |
| QQP - Dev acc/F1 | 91.4/88.4 | 10% | 79.8/65.0 | 88.1/82.8 | 89.7/86.2 | **90.2/86.8** |
|  |  | 3% | 72.4/57.8 | 87.0/81.9 | 86.1/81.5 | **89.1/85.5** |

teacher. The training objective is a convex combination of the training loss and a knowledge distillation loss on the output distributions. Figure 3 shows the results on SQuAD, MNLI, and QQP for the three pruning methods boosted with distillation. Overall, we observe that the relative comparisons of the pruning methods remain unchanged while the performances are strictly increased. Table 3 shows for instance that on SQuAD, movement pruning at 10% goes from 81.7 F1 to 84.3 F1. When combined with distillation, soft movement pruning yields the strongest performances across all pruning methods and studied datasets: it reaches 95% of BERT-base with only a fraction of the weights in the encoder (∼5% on SQuAD and MNLI).

# 7    Analysis

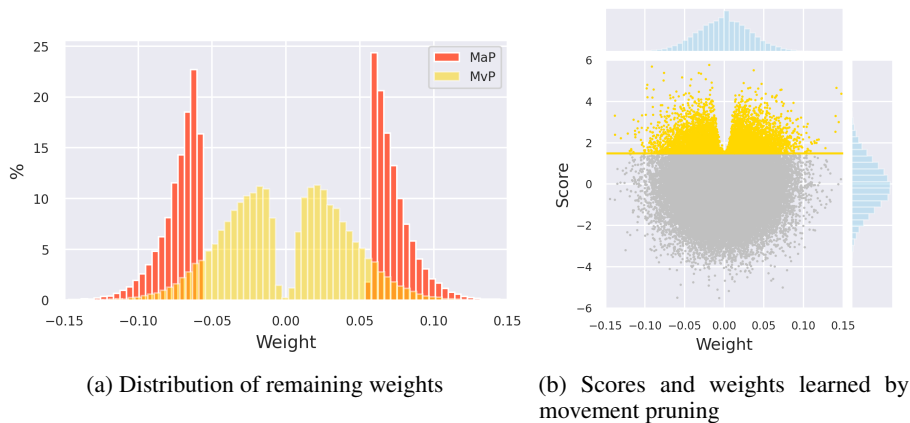

(a) Distribution of remaining weights

(b) Scores and weights learned by movement pruning

Figure 4: Magnitude pruning and movement pruning leads to pruned models with radically different weight distribution.

Figure 5: Comparison of local and global selections of weights on SQuAD at different sparsity levels. **For magnitude and movement pruning, local and global Top$_v$ performs similarly at all levels of sparsity.**

Figure 6: **Remaining weights per layer in the Transformer.** Global magnitude pruning tends to prune uniformly layers. Global 1st order methods allocate the weight to the lower layers while heavily pruning the highest layers.

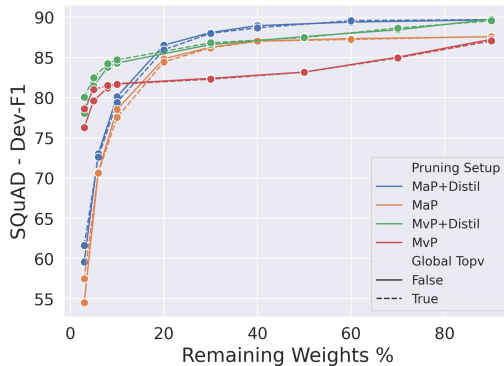

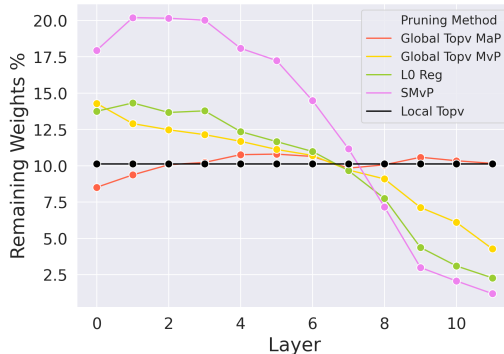

**Movement pruning is adaptive** Figure 4a compares the distribution of the remaining weights for the same matrix of a model pruned at the same sparsity using magnitude and movement pruning. We observe that by definition, magnitude pruning removes all the weights that are close to zero, ending up with two clusters. In contrast, movement pruning leads to a smoother distribution, which covers the whole interval except for values close to 0.

Figure 4b displays each individual weight against its associated importance score in movement pruning. We plot pruned weights in grey. We observe that movement pruning induces no simple relationship between the scores and the weights. Both weights with high absolute value or low absolute value can be considered important. However, high scores are systematically associated with non-zero weights (and thus the "v-shape"). This is coherent with the interpretation we gave to the scores (section 4): a high score **S** indicates that during fine-tuning, the associated weight moved away from 0 and is thus non-null.

**Local and global masks perform similarly** We study the influence of the locality of the pruning decision. While local Top$_v$ selects the $v\%$ most important weights matrix by matrix, global Top$_v$ uncovers non-uniform sparsity patterns in the network by selecting the $v\%$ most important weights in the whole network. Previous work has shown that a non-uniform sparsity across layers is crucial to the performance in high sparsity regimes [He et al., 2018]. In particular, Mallya and Lazebnik [2018] found that the sparsity tends to increase with the depth of the network layer.

Figure 5 compares the performance of local selection (matrix by matrix) against global selection (all the matrices) for magnitude pruning and movement pruning. Despite being able to find a global sparsity structure, we found that global did not significantly outperform local, except in high sparsity regimes (2.3 F1 points of difference with 3% of remaining weights for movement pruning). Even though the distillation signal boosts the performance of pruned models, the end performance difference between local and global selections remains marginal.

Figure 6 shows the remaining weights percentage obtained per layer when the model is pruned until 10% with global pruning methods. Global weight pruning is able to allocate sparsity non-uniformly through the network, and it has been shown to be crucial for the performance in high sparsity regimes [He et al., 2018]. We notice that except for global magnitude pruning, all the global pruning methods tend to allocate a significant part of the weights to the lowest layers while heavily pruning in the highest layers. Global magnitude pruning tends to prune similarly to local magnitude pruning, i.e., uniformly across layers.

## 8 Conclusion

We consider the case of pruning of pretrained models for task-specific fine-tuning and compare zeroth- and first-order pruning methods. We show that a simple method for weight pruning based on straight-through gradients is effective for this task and that it adapts using a first-order importance score. We apply this movement pruning to a transformer-based architecture and empirically show that our method consistently yields strong improvements over existing methods in high-sparsity regimes. The analysis demonstrates how this approach adapts to the fine-tuning regime in a way that magnitude pruning cannot. In future work, it would also be interesting to leverage group-sparsity inducing penalties [Bach et al., 2011] to remove entire columns or filters. In this setup, we would associate a score to a group of weights (a column or a row for instance). In the transformer architecture, it would give a systematic way to perform feature selection and remove entire columns of the embedding matrix.

## 9 Broader Impact

This work is part of a broader line of research on reducing the memory size of state-of-the-art models in Natural Language Processing (and more generally in Artificial Intelligence). This line of research has potential positive impact in society from a privacy and security perspective: being able to store and run state-of-the-art NLP capabilities on devices (such as smartphones) would erase the need to send API calls (with potentially private data) to a remote server. It is particularly important since there is a rising concern about the potential negative uses of centralized personal data. Moreover, this is complementary to hardware manufacturers' efforts to build chips that will considerably speedup inference for sparse networks while reducing the energy consumption of such networks.

From an accessibility standpoint, this line of research has the potential to give access to extremely large models [Raffel et al., 2019, Brown et al., 2020] to the broader community, and not only big labs with large clusters. Extremely compressed models with comparable performance enable smaller teams or individual researchers to experiment with large models on a single GPU. For instance, it would enable the broader community to engage in analyzing a model's biases such as gender bias [Lu et al., 2018, Vig et al., 2020], or a model's lack of robustness to adversarial attacks [Wallace et al., 2019]. More in-depth studies are necessary in these areas to fully understand the risks associated to a model and create robust ways to mitigate them before massively deploying these capabilities.

## Acknowledgments and Disclosure of Funding

This work is conducted as part the authors' employment at Hugging Face.

## Footnotes

[1]We also experimented with $\sum_{i,j}|S_{i,j}|$ but it turned out to be harder to tune while giving similar results.

[2]Preliminary experiments showed that increasing the number of pruning steps tended to improve the end performance

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
