[Supplementary Material]

# A Appendices

## A.1 Guarantees on the decrease of the training loss

As the scores are updated, the relative order of the importances is likely shuffled, and some connections will be replaced by more important ones. Under certain conditions, we are able to formally prove that as these replacements happen, the training loss is guaranteed to decrease. Our proof is adapted from [Ramanujan et al., 2020] to consider the case of fine-tuable $\mathbf{W}$.

We suppose that (a) the training loss $\mathcal{L}$ is smooth and admits a first-order Taylor development everywhere it is defined and (b) the learning rate of $\mathbf{W}$ ($\alpha_{\mathbf{W}} > 0$) is small. We define the TopK function as the analog of the $\mathrm{Top}_v$ function, where $k$ is an integer instead of a proportion. We first consider the case where $k = 1$ in the TopK masking, meaning that only one connection is remaining (and the other weights are deactivated/masked). Let's denote $W_{i,j}$ this sole remaining connection at step $t$. Following Eq (1), it means that $\forall 1 \leq u, v \leq n, S_{u,v}^{(t)} \leq S_{i,j}^{(t)}$.

We suppose that at step $t + 1$, connections are swapped and the only remaining connection at step $t + 1$ is $(k, l)$. We have:

$$\begin{cases} \text{At } t, & \forall 1 \leq u, v \leq n, S_{u,v}^{(t)} \leq S_{i,j}^{(t)} \\ \text{At } t + 1, & \forall 1 \leq u, v \leq n, S_{u,v}^{(t+1)} \leq S_{k,l}^{(t+1)} \end{cases} \tag{6}$$

Eq (6) gives the following inequality: $S_{k,l}^{(t+1)} - S_{k,l}^{(t)} \geq S_{i,j}^{(t+1)} - S_{i,j}^{(t)}$. After re-injecting the gradient update in Eq (2), we have:

$$-\alpha_{\mathbf{S}} \frac{\partial \mathcal{L}}{\partial a_k} W_{k,l}^{(t)} x_l \geq -\alpha_{\mathbf{S}} \frac{\partial \mathcal{L}}{\partial a_i} W_{i,j}^{(t)} x_j \tag{7}$$

Moreover, the conditions in Eq (6) lead to the following inferences: $a_i^{(t)} = W_{i,j}^{(t)} x_j$ and $a_k^{(t+1)} = W_{k,l}^{(t+1)} x_l$.

Since $\alpha_{\mathbf{W}}$ is small, $||(a_i^{(t+1)}, a_k^{(t+1)}) - (a_i^{(t)}, a_k^{(t)})||_2$ is also small. Because the training loss $\mathcal{L}$ is smooth, we can write the 1st order Taylor development of $\mathcal{L}$ in point $(a_i^{(t)}, a_k^{(t)})$:

$$\begin{aligned} \mathcal{L}&(a_i^{(t+1)}, a_k^{(t+1)}) - \mathcal{L}(a_i^{(t)}, a_k^{(t)}) \\ &\approx \frac{\partial \mathcal{L}}{\partial a_k}(a_k^{(t+1)} - a_k^{(t)}) + \frac{\partial \mathcal{L}}{\partial a_i}(a_i^{(t+1)} - a_i^{(t)}) \\ &= \frac{\partial \mathcal{L}}{\partial a_k} W_{k,l}^{(t+1)} x_l - \frac{\partial \mathcal{L}}{\partial a_i} W_{i,j}^{(t)} x_j \\ &= \frac{\partial \mathcal{L}}{\partial a_k} W_{k,l}^{(t+1)} x_l + (-\frac{\partial \mathcal{L}}{\partial a_k} W_{k,l}^{(t)} x_l + \frac{\partial \mathcal{L}}{\partial a_k} W_{k,l}^{(t)} x_l) - \frac{\partial \mathcal{L}}{\partial a_i} W_{i,j}^{(t)} x_j \\ &= \frac{\partial \mathcal{L}}{\partial a_k}(W_{k,l}^{(t+1)} x_l - W_{k,l}^{(t)} x_l) + (\frac{\partial \mathcal{L}}{\partial a_k} W_{k,l}^{(t)} x_l - \frac{\partial \mathcal{L}}{\partial a_i} W_{i,j}^{(t)} x_j) \\ &= \frac{\partial \mathcal{L}}{\partial a_k} x_l (-\alpha_{\mathbf{W}} \frac{\partial \mathcal{L}}{\partial a_k} x_l m(S^{(t)})_{k,l}) + (\frac{\partial \mathcal{L}}{\partial a_k} W_{k,l}^{(t)} x_l - \frac{\partial \mathcal{L}}{\partial a_i} W_{i,j}^{(t)} x_j) \end{aligned} \tag{8}$$

The first term is null because of inequalities (6) and the second term is negative because of inequality (7). Thus $\mathcal{L}(a_i^{(t+1)}, a_k^{(t+1)}) \leq \mathcal{L}(a_i^{(t)}, a_k^{(t)})$: when connection $(k, l)$ becomes more important than $(i, j)$, the connections are swapped and the training loss decreases between step $t$ and $t + 1$.

Similarly, we can generalize the proof to a set $\mathcal{E} = \{((a_i, b_i), (c_i, d_i)); i \leq N\}$ of $N$ swapping connections.

We note that this proof is not specific to the *TopK* masking function. In fact, we can extend the proof using the *Threshold* masking function $\mathbf{M} := (\mathbf{S} >= \tau)$ [Mallya and Lazebnik, 2018]. Inequalities (6) are still valid and the proof stays unchanged.

Last, we note these guarantees do not hold if we consider the absolute value of the scores $|S_{i,j}|$ (as it is done in Ding et al. [2019] for instance). We prove it by contradiction. If it was the case, it would also be true one specific case: the *negative threshold* masking function ($\mathbf{M} := (\mathbf{S} < \tau)$ where $\tau < 0$).

We suppose that at step $t+1$, the only remaining connection $(i,j)$ is replaced by $(k,l)$:

$$
\begin{cases}
\text{At } t, & \forall 1 \leq u, v \leq n, S_{i,j}^{(t)} \leq \tau \leq S_{u,v}^{(t)} \\
\text{At } t+1, & \forall 1 \leq u, v \leq n, S_{k,l}^{(t+1)} \leq \tau \leq S_{u,v}^{(t+1)}
\end{cases}
\tag{9}
$$

The inequality on the gradient update becomes: $-\alpha_{\mathbf{S}} \frac{\partial \mathcal{L}}{\partial a_k} W_{k,l}^{(t)} x_l < -\alpha_{\mathbf{S}} \frac{\partial \mathcal{L}}{\partial a_i} W_{i,j} x_j$ and following the same development as in Eq (8), we have $\mathcal{L}(a_i^{(t+1)}, a_k^{(t+1)}) - \mathcal{L}(a_i^{(t)}, a_k^{(t)}) \geq 0$: the loss increases. We proved by contradiction that the guarantees on the decrease of the loss do not hold if we consider the absolute value of the score as a proxy for importance.

## A.2   Code and Hyperparameters

Our code to reproduce our results along with the commands to launch the scripts are available[3] through the Hugging Face Transformers library [Wolf et al., 2019]. We also detail all the hyperparameters used in our experiments.

All of the presented experiments run on a single 16GB V100.

## A.3   Inference speed

Early experiments show that even though models fine-pruned with movement pruning are extremely sparse and can be stored efficiently, they do not benefit from significant improvement in terms of inference speed when using the standard PyTorch inference. In fact, in our implementation, sparse matrices are simply matrices with lots of 0 inside which does not induce significant inference speedup without specific hardware optimizations. Recently, hardware manufacturers have announced chips specifically designed for sparse networks[4].

## Footnotes

[3]huggingface.co/mvp

[4]https://devblogs.nvidia.com/nvidia-ampere-architecture-in-depth/