[Reviews · NeurIPS 2020]

Review 1

Summary and Contributions: The paper proposes movement pruning, which is a first order weight pruning method that allows pruning to be more easily adaptive during fine tuning. This is compared to magnitude pruning, which has been effective for supervised learning settings. However, an advantage of movement pruning is that when the weights are shifting during fine tuning, movement pruning is more adaptive to this scenario.

Strengths: - nice explanation of the technique, particularly in section 4 - very nice figures displaying the technique and comparing it to magnitude pruning - strong performance on GLUE benchmark tasks and squad

Weaknesses: - there are very few experimental details about distillation - is this distillation only on the training set, or is there data augmentation? - it is difficult to understand e.g. figure 5, there are a lot of lines on top of each other - the main metrics reported are performance compared to remaining weights, but the authors could report flops or model size, to make this much more concrete

Correctness: overall looks good

Clarity: paper was clear and well written

Relation to Prior Work: yes, related works section looks good

Reproducibility: No

Additional Feedback: - can the authors comment how movement pruning might work for generative tasks, for example if T5 or BART were finetuned? thanks authors for the review response.


Review 2

Summary and Contributions: After author response: Thanks for clarifying that the reported sparsity is relative to BERT base. This makes the reported comparisons more fair than I originally realized. I've updated my scores accordingly. === This paper proposes an approach for pruning pretrained language models during the fine-tuning process. Compared to past work that prunes weights based on their magnitude, the authors instead propose to prune based on the degree to which weights move toward 0 during fine-tuning. They evaluate their approach on several NLP tasks and compare to several pruning baselines and other parameter reduction techniques (LayerDrop, MiniBERT). The proposed approach shows particularly strong performance relative to baselines in the high sparsity regime (>95% sparsity), although there is still a gap with the original unpruned model. The authors show that this gap can be further reduced with knowledge distillation.

Strengths: - The proposed approach is well motivated, straightforward to implement and performs well in high sparsity settings. - The evaluation appears solid and the proposed approach is compared to strong baselines. - There is decent theoretical and empirical analysis confirming that the proposed approach does what it claims to (prune weights moving toward 0).

Weaknesses: - While the baselines are strong, the way they are reported may be a bit misleading. In particular, models are compared based on the sparsity percentage, which puts models with fewer parameters (e.g., MiniBERT) at a disadvantage. - As with most work on pruning, it is not yet possible to realize efficiency gains on GPU.

Correctness: The claims are mostly well supported, although the comparison to non-sparse models with fewer parameters (e.g., MiniBERT, possibly LayerDrop) are not quite fair since they are based on relative sparsity percentage instead of total non-zero parameter count.

Clarity: The paper is very clearly written.

Relation to Prior Work: The comparison to related work seems quite comprehensive and I did not find any missing related work.

Reproducibility: Yes

Additional Feedback: - For the results in Figure 2, what does the x-axis represent for models with different numbers of parameters? For example, if a MiniBERT model has half as many parameters as BERT-base, then comparing "10% remaining weights" seems a bit unfair. What would the figure look like if the x-axis were instead the number of non-zero parameters? - You evaluate on one span extraction and two paired sentence classification tasks, but no single sentence classification tasks. Why not replace one of the sentence pair tasks with SST-2, for example? I expect the results would be similar, but it would make the experiments a bit more compelling. Presentation suggestions: - In Figure 3, it's a bit hard to compare the results to those in Figure 2. Perhaps consider plotting the delta F1 or accuracy instead of absolute values. - The presentation in Section 4 is a bit jarring, because you switch from discussing movement pruning to L_0 back to movement pruning. It may be clearer to swap the L_0 and Method Interpretation paragraphs. - Could you please add the raw values from Figures 2/3 to the appendix? - In Figure 5, instead of having "Global Topv; False; True" in the legend, perhaps change to "Topv; Local; Global" - typo (line 115): "yield a similar update $L_0$" -> "yield a similar update as $L_0$" - typo (line 265): stray sentence fragment: "Privacy-preserving Machine Learning ."


Review 3

Summary and Contributions: In this paper the authors propose a method for pruning neural network weights based on first-order statistics rather than 0th-order statistics (e.g. the common magnitude-based pruning), which they claim should work better in the transfer learning setting, i.e.fine-tuning a pre-trained language model. They show that their approach leads to better accuracy/F1 at a given sparsity over various baselines when used during fine-tuning on SQuAD, MNLI and QQP. Following discussion: The author response didn't change my opinion enough for me to increase my score (and they did not address my concerns with one of the related works I mentioned, maybe it's less relevant than I thought, but they had room to explain that), and I agree with R4 that I would like to see more analysis of what is happening at low sparsity levels.

Strengths: - well-motivated and effective (seemingly good empirical results) technique for task-specific compression of fine-tuned pre-trained language models, an important area of research to make these models usable in "real", deployed scenarios, and more accessible generally - A nice analysis of results

Weaknesses: - At least one highly related work is missing (see below) - empirical results don't compare to any other published work, only self-implemented baselines. I'm not convinced that there exist no other relevant works on pruning pre-trained LMs that could be compared to. I could definitely be convinced that this is acceptable given existing work, but I'd like to even see an explanation of why existing work isn't comparable in the paper.

Correctness: The results presented seem correct, but I'm concerned about the lack of comparison to other approaches for compressing LMs during fine-tuning. For example, https://arxiv.org/abs/1909.12486 seems comparable, also https://arxiv.org/abs/2002.08307, https://arxiv.org/abs/1910.06360, and https://arxiv.org/abs/2002.11794

Clarity: The paper is very well written and easy to follow. There are not enough details on optimization included in the main paper (I did not read the appendix) to replicate this work.

Relation to Prior Work: A highly related work from NeurIPS 2019 is missing: Sparse Networks from Scratch: Faster Training without Losing Performance (https://arxiv.org/abs/1907.04840). It may even make sense to compare to this work as a baseline. See also papers under "Correctness" (I think some of these are included and some are not.).

Reproducibility: No

Additional Feedback: - you may want to consider reformatting table 2, it is quite hard to parse - similarly, Figure 2 is quite busy and hard to read; you may want to remove some of the lines from this figure, and include the full plots as supplemental material


Review 4

Summary and Contributions: This paper shows that the magnitude pruning methods are less effective in the case of pruning of pre-trained models for task-specific fine-tuning, and proposed a simple pruning approach, movement pruning (mvp), which is first-order and more adaptive. Rather than pruning weights with small absolute values like magnitude pruning, mvp prunes weights that are shrinking towards 0. The authors conducted experiments on BERT fine-tuning, which show that mvp outperforms magnitude pruning method and other 1st order methods at high sparsity level.

Strengths: This paper is well-written. The proposed approachs and conducted experiments are clearly described. The visualization and analysis about the differences between magnitude pruning and movement pruning (in particular, Figure 1 and Figure 4) in the paper is very intuitive and intriguing. For example, they found the distribution of remaining weights after pruning with movement pruning method smoother than that with magnitude pruning. Moreover, the weights that are close to 0 are also less important, which is consistent with magnitude pruning. The proposed approach is simple and effective, compared with the similar 1st-order method, L0 regularization. Besides, the experimental results demonstrate that the approach (and its soft version) works well at high sparsity level.

Weaknesses: The designed movement pruning approach is somehow lack of novelty, as various pruning heuristics (based on activations, redundancy, second derivatives, channels, etc.) have been proposed. Besides, the experimental results show that the movement pruning method performs worse than magnitude pruning method when at low sparsity level but the authors did not explain why. Does the poor performance at low sparsity level mean that the proposed importance criterion is not suitable for low sparsity pruning?

Correctness: The claims, method, and empirical methodology in the paper are correct.

Clarity: Overall, the paper is well-written. The method and experiments are clearly described. Also, the figures and tables in the paper well expressed the relationship and difference between their method and the related methods. It would be better to make Section 4 clearer, especially the part about L0 regularization.

Relation to Prior Work: Yes, the authors compare their proposed method with magnitude pruning and L0 regularization. The characteristics of these methods are highlighted in Table 1. The difference between movement pruning and magnitude pruning is depicted in Figure 1 and Figure 4.

Reproducibility: Yes

Additional Feedback: More analysis about the reasons why movement pruning performs worse than magnitude pruning at low sparsity level is suggested. As shown in Figure 4, the movement pruning is more general and adaptive than magnitude pruning, but the results at low sparsity level show that simply removing weights with small absolute values yields better performance. That means, when removing a small proportion of connections, the distance from 0 is a better importance criterion than movement. But when removing a large proportion of connections, some important weights but with relatively small absolute values are wrongly removed by magnitude pruning. Can you give some intuitive explanations or deeper analysis about this phenomenon?

[Author Response · NeurIPS 2020]

We sincerely thank the reviewers for sharing their valuable feedback while pointing out weaknesses in our work and suggesting presentations improvements.

**All - Report the model size as opposed to sparsity percentage / Claims are not quite fair since they are based on relative sparsity percentage instead of total non-zero parameter count.** There was some confusion, which we will clarify, in R1/R2 about reporting sparsity percentages. All percentages are relative to BERT base, and correspond *exactly* to model size (even for MiniBERT and Layer Drop). To address we will include in the appendix the main results (Figures 2&3, Tables 2&3) to plot the performance against the number of non-zero parameters in the encoder. For instance, 3% corresponds to 2.6 millions (M) non-zero parameters in the encoder, 10% to 8.5M, 20% to 17M.

**R1 - Is this distillation only on the training set, or is there data augmentation?** We do not use data augmentation in any of our experiments. The model is trained solely on the training set. The distillation signal comes from the dense teacher of the same size plotted in Figures 2&3 in cyan. We follow the vanilla setup described in Hinton et al. [2014].

**R1 - Can the authors comment how movement pruning might work for generative tasks?** Interesting idea. For encoder-decoder setups, we can augment the Fully Connected layers in the Transformer block with score matrices, learn these scores during training and discard them after pruning. While we have not yet tested this extensively, initial small scale experiments on DistilBART for summarization (3 layers encoder and 3 layers decoder) give the following results at 90% sparsity (Rouge-2/Rouge-L on XSum): Dense=12.3/27.3, MaP=8.8/22.3, $L_0$=9.8/23.6, MvP=11.0/25.0, indicating that movement pruning is also promising in this setting.

**R2 - As with most work on pruning, it is not yet possible to realize efficiency gains on GPU.** We agree that inference speed for pruned models is still an open concern. However, we argue that our work (and other pruning studies) have direct applications in real-world settings. As highlighted in Han et al. [2016], most of the energy consumption for on-device deep learning comes from the loading of the weights. Reducing the memory size of the model is a crucial step towards enabling more on-device applications even without speed-ups. Moreover, chip manufacturers are making progress towards accelerating sparse models in many settings (for instance, A100 from Nvidia). For our models, working with new sparse inference frameworks, we are already able to get 3x speed gain on CPU using a sparse model.

**R3 - The results presented seem correct, but I'm concerned about the lack of comparison to other approaches for compressing LMs during fine-tuning.** The reviewer mentions several specific papers. We have compared against (1909.12486) in our submission: it is displayed as *RPP* in Figure 2&4. Works (2002.08307) and (2002.11794) apply unstructured magnitude pruning as a post-hoc operation whereas we use *automated gradual pruning* [Zhu and Gupta, 2018], a variant of magnitude pruning which improves on these methods by enabling masked weights to be updated. For instance, (2002.08307) obtains a score of 58.7 on MNLI compared to 78.4 at 90% sparsity with automated gradual pruning. Finally (1910.06360) compares multiple methods to compute structured masking ($L_0$ regularization and head importance as described in [Michel et al., 2019]) and found that structured $L_0$ regularization performs best. We did not find any implementation for this work, so to be fair, we presented a strong unstructured $L_0$ regularization baseline. We will also add a reference to the related NeurIPS2019 work (1907.04840).

**R4 - The designed movement pruning approach is lacking of novelty, as various pruning heuristics have been proposed.** As highlighted in Section 4, our method is indeed similar to previous general propositions, such as $L_0$ regularization. We frame our study in the context of *transfer learning* and how it differs from standard supervised learning. In this setting, the change paradigm (moving away from 0 instead of being far from 0) is crucial in high sparsity regimes. To the best of our knowledge, it is not a perspective that is commonly developed in other works since a significant part of these focus on pruning non-pre-trained models. Movement pruning shows strong performances in this context, out-performing $L_0$ regularization while being very simple (both to understand and implement).

**R4 - Does the poor performance at low sparsity level mean that the proposed importance criterion is not suitable for low sparsity pruning?** We have not found a convincing explanation for this phenomenon: movement-based pruning compare less favorably against magnitude pruning at low sparsity. We leave this interesting exploration for future work.

# References

Geoffrey E. Hinton, Oriol Vinyals, and Jeffrey Dean. Distilling the knowledge in a neural network. In *NIPS*, 2014.

Song Han, Xingyu Liu, Huizi Mao, Jing Pu, Ardavan Pedram, Mark Horowitz, and William J. Dally. Eie: Efficient inference engine on compressed deep neural network. In *ISCA*, 2016.

Michael Zhu and Suyog Gupta. To prune, or not to prune: exploring the efficacy of pruning for model compression. In *ICLR*, 2018.

Paul Michel, Omer Levy, and Graham Neubig. Are sixteen heads really better than one? In *NeurIPS*, 2019.


[Meta-Review · NeurIPS 2020]

This paper proposes movement pruning - a first order weight pruning method that allows pruning to be more easily adaptive during fine tuning. This is compared to traditional magnitude pruning. Movement pruning is shown to be more adaptive for the scenario where the weights are shifting during fine tuning. All four reviewers recommend accepting this paper (though some found it borderline). I agree with the reviewers and recommend acceptance. One weakness pointed out is that while the baselines are strong, the way they are reported may be a bit misleading. In particular, models are compared based on the sparsity percentage, which puts models with fewer parameters (e.g., MiniBERT) at a disadvantage. The clarification that the authors reported sparsity relative to BERT base (rather than relative sparsity) clarified that the comparison seems more fair than originally realized. I encourage authors to take into account the reviewers' suggestions in the final version. In particular, they should add the missing citations pointed out by R3, and clarify the relation with prior work in a discussion section.